# The Surgical Imprint: How Operative Trauma May Shape Radiation Tolerance After Prostatectomy

**DOI:** 10.3390/cancers17162685

**Published:** 2025-08-18

**Authors:** Alessio G. Morganti, Gabriella Macchia, Filippo Mammini, Arina A. Zamfir, Milly Buwenge, Francesco Cellini, Lorenzo Bianchi, Riccardo Schiavina, Eugenio Brunocilla, Francesco Deodato, Savino Cilla

**Affiliations:** 1Radiation Oncology, Department of Medical and Surgical Sciences (DIMEC), Alma Mater Studiorum—Bologna University, 40138 Bologna, Italy; alessio.morganti2@unibo.it (A.G.M.); filippo.mammini@studio.unibo.it (F.M.); milly.buwenge2@unibo.it (M.B.); 2Radiation Oncology, IRCCS Azienda Ospedaliero—Universitaria di Bologna. 40138 Bologna, Italy; 3Radiotherapy Unit, Responsible Research Hospital, 86100 Campobasso, Italy; gabriella.macchia@responsible.hospital (G.M.); francesco.deodato@unicatt.it (F.D.); 4Dipartimento Universitario Diagnostica per Immagini, Radioterapia Oncologica ed Ematologia, Università Cattolica del Sacro Cuore, 00168 Rome, Italy; francesco.cellini@policlinicogemelli.it; 5Dipartimento di Diagnostica per Immagini, Radioterapia Oncologica ed Ematologia, Fondazione Policlinico Universitario “A. Gemelli” IRCCS, 00168 Rome, Italy; 6Division of Urology, IRCCS Azienza Ospedaliero—Universitaria di Bologna, 40138 Bologna, Italy; lorenzo.bianchi13@unibo.it (L.B.); riccardo.schiavina3@unibo.it (R.S.); eugenio.brunocilla@unibo.it (E.B.); 7Department of Medical and Surgical Sciences (DIMEC), Alma Mater Studiorum—Bologna University, 40138 Bologna, Italy; 8Istituto di Radiologia, Università Cattolica del Sacro Cuore, 00168 Roma, Italy; 9Medical Physics Unit, Responsible Research Hospital, 86100 Campobasso, Italy; savino.cilla@responsible.hospital

**Keywords:** post-prostatectomy radiotherapy, acute toxicity, genitourinary toxicity, gastrointestinal toxicity, minimally invasive surgery, open prostatectomy, surgical trauma, radiosensitivity, CTV–PTV margin, salvage radiotherapy

## Abstract

Radiotherapy is often used after prostate surgery when blood tests suggest that the cancer may have come back. In our recent study, we found something unexpected: men who had minimally invasive prostate surgery (such as robotic or laparoscopic approaches) experienced fewer short-term side effects from radiotherapy than those who had traditional open surgery. This difference could not be explained by the radiation dose or technique. In this article, we explore possible biological reasons for this finding. We suggest that open surgery may cause more inflammation and scarring, which could make tissues more sensitive to radiation. Understanding this interaction may help doctors personalize treatment plans and reduce side effects for future patients. This article is a Perspective paper that expands upon the ICAROS study that we recently published in *Cancers*; it therefore revisits the same dataset to explore its clinical and biological implications rather than presenting new patient data.

## 1. Introduction

Post-prostatectomy radiotherapy (RT) is a standard treatment option for patients with adverse pathological findings or biochemical relapse after radical prostatectomy. Contemporary guidelines from EAU-EANM-ESTRO-ESUR-ISUP-SIOG [1], ASTRO [2], and NCCN [3] endorse both adjuvant and early-salvage RT, noting oncological benefits in biochemical control and metastasis-free survival [4,5]. Yet, despite modern image-guided and intensity-modulated techniques, acute grade ≥ 2 genitourinary (GU) and gastrointestinal (GI) toxicities still occur in up to one-third of treated patients, with significant impacts on quality of life [6].

Traditionally, variability in post-prostatectomy toxicity has been attributed to radiation-related factors: total dose, fractionation schedule, margins from clinical target volume (CTV) to planning target volume (PTV), and daily image guidance. Randomized clinical trials (RCT) and large cohort studies (e.g., GETUG-AFU 17 [7] and RADICALS-RT [8]) have clarified the dose–toxicity relationship and other studies [9,10,11] supported margin reduction when robust targeting was available, but inter-study heterogeneity in acute events remains high. This inconsistency suggests that variables outside the radiation suite modulate tissue radiosensitivity.

Open (retropubic) prostatectomy is consistently associated with a markedly higher peri-operative morbidity than minimally invasive techniques. In the Swiss nationwide registry of 8593 procedures, the overall Clavien complication rate reached 17.1% after open surgery but only 9–10% after laparoscopic or robotic approaches [12]. A recent meta-analysis pooling five RCTs or high-quality comparative studies confirmed that robot-assisted surgery reduced estimated blood loss, transfusion requirement (odds ratio 0.17, *p* < 0.001) and overall complications (odds ratio 0.71, *p* = 0.001) relative to the open approach [13].

In a recent multicenter analysis of 454 men undergoing salvage RT, we applied machine-learning models to identify predictors of acute GU and GI toxicity. Unexpectedly, and without any a priori hypothesis, the type of prostatectomy (open vs. minimally invasive) emerged as the most powerful and consistent predictor of ≥grade 2 toxicity, even when controlling for dose, margins, nodal irradiation and beam delivery technique [14]. This result was surprising and has not been previously reported.

The current paper arises directly from that observation. Here, we seek to explore possible biological and technical mechanisms that could explain this apparent protective effect of minimally invasive surgery. Drawing on the current understanding of surgical trauma, RT response, and pelvic tissue biology, we propose a model in which the “surgical imprint” alters the microenvironment of the irradiated tissues, thereby conditioning their vulnerability to radiation-induced damage. While our findings are hypothesis-generating, they open a novel area of investigation at the intersection of surgery and radiation oncology. Therefore, the objective of the present Perspective is interpretative: we synthesize the biological effects of surgery and RT to propose a conceptual framework that explains why surgical access emerged as the dominant predictor of acute toxicity. No new patients were accrued and no additional statistical analyses were undertaken beyond graphical reformats intended for a clinical audience.

## 2. Complication Profile of Open Versus Minimally Invasive Prostatectomy

### 2.1. Contemporary Comparative Data

Minimally invasive radical prostatectomy, particularly in its robot-assisted form (RARP), has increasingly replaced the open retropubic approach (ORP) in the surgical treatment of localized prostate cancer. A growing body of high-level evidence demonstrates that RARP offers significant reductions in surgical morbidity without compromising oncologic safety. A prospective meta-analysis of 21 studies [15] reported that, despite slightly longer operative times, RARP results in substantially lower blood loss (mean reduction of approximately 500 mL), a more than 75% lower transfusion rate (OR 0.23), a 39% reduction in overall complications (OR 0.61), and a significantly shorter hospital stay (mean difference of 1.59 days). These findings are consistent with those of the Cochrane review [16], which also highlighted reduced postoperative pain during the first week after minimally invasive surgery.

RCTs have confirmed these perioperative advantages. In the Brazilian single-center RCT [17], RARP was associated with a median blood loss of 250 mL compared to 720 mL for ORP, and although the difference in complication rates (11.1% vs. 17.3%) did not reach statistical significance, RARP demonstrated superior functional outcomes: continence recovery at 18 months was 95.4% versus 78.8%, and erectile function recovery was significantly higher within the first year. Similar reductions in blood loss were observed in the Australian RCT [18], which reported comparable urinary and sexual function outcomes and a significantly lower biochemical recurrence rate in the RARP group at 24 months.

Longer-term data support the durability of these benefits. The Swedish multicenter controlled cohort study [19] followed 4003 patients for 8 years and reported equivalent urinary incontinence rates, but a modest reduction in erectile dysfunction and prostate cancer–specific mortality in the RARP group, especially among high-risk patients. These findings suggest that the short-term functional benefits of RARP are not offset by inferior oncological outcomes.

A systematic overview of reviews [20] underscores that, although the widespread adoption of robotic surgery has outpaced the availability of high-certainty evidence, for radical prostatectomy there is consistent documentation of improved early recovery and lower rates of biochemical recurrence when compared to ORP. Taken together, current data indicate that minimally invasive approaches, when performed by experienced surgeons, reduce perioperative blood loss, transfusion requirements, early postoperative pain, and mild-to-moderate complications, while facilitating faster recovery of urinary and sexual function. Importantly, these benefits are achieved without sacrificing long-term oncological efficacy, though future studies should further investigate outcomes beyond a 10-year horizon and address cost effectiveness.

### 2.2. Tissue-Level Effects of Surgical Approach in Radical Prostatectomy

#### 2.2.1. Surgical Trauma and Local Microenvironment Changes

Surgical removal of the prostate induces an acute perturbation in the local tissue microenvironment. This response is characterized by a rapid increase in pro-inflammatory cytokines, most notably interleukin-6 (IL-6), along with IL-8/CXCL8, growth factors, and matrix remodelers. A 2023 study profiling perioperative cytokines in prostate cancer patients confirmed a sharp rise in IL-6 immediately after tumor resection, while noting a transient drop in transforming growth factor β1 (TGF-β1), possibly due to the elimination of tumor-derived sources, with subsequent normalization in 1–2 weeks [21].

As the wound-healing process advances, TGF-β signaling becomes dominant, orchestrating fibroblast activation and extracellular matrix deposition, key drivers of fibrotic remodeling in the surgical bed. In parallel, hypoxia arises due to vessel ligation and tissue injury, stabilizing hypoxia-inducible factor-1α (HIF-1α), which further promotes angiogenesis and fibrogenesis.

The magnitude of these responses is influenced by surgical technique. Minimally invasive prostatectomy elicits a weaker inflammatory cascade and less immune disturbance than ORP. A 2020 review emphasized that minimally invasive prostatectomy induces less immunosuppression and systemic stress than ORP [22]. Patients undergoing robotic or laparoscopic surgery exhibit lower postoperative IL-6 and C-reactive protein (CRP) peaks, indicating reduced systemic inflammation [23].

In conclusion, perioperative studies reported lower early IL-6 and CRP peaks and smaller lactate increases after robotic versus open prostatectomy, despite slightly longer operating time, supporting the biological plausibility of a more favorable post-surgical microenvironment.

#### 2.2.2. Cytokine Signaling and Fibrotic Response Post-Radiotherapy

Pelvic RT further amplifies inflammatory pathways. IL-6 levels increase significantly during the course of RT, and higher baseline levels of TGF-β1 and IL-6 have been correlated with more severe acute GU toxicity [24].

Radiation damage also triggers the release of pro-inflammatory mediators such as IL-1β and TNF-α, which activate fibroblasts and promote chronic fibrosis through upregulation of TGF-β. This cytokine cascade exacerbates local tissue injury and contributes to both acute symptoms and long-term complications such as urethral strictures and rectal fibrosis [25].

Thus, a surgically pre-conditioned environment already rich in IL-6 and early TGF-β signaling is likely to respond more aggressively to radiation injury, explaining the heightened toxicity observed in patients with prior open surgery.

#### 2.2.3. Systemic Immune Modulation: Open vs. Minimally Invasive Surgery

Beyond local effects, surgical trauma impacts systemic immunity. ORP is associated with a more pronounced postoperative rise in myeloid-derived suppressor cells and delayed recovery of natural killer (NK) and CD8+ T-cell populations [23]. In contrast, minimally invasive prostatectomy confers an “immunity-sparing” profile: lower IL-6 and IL-10 levels, a more favorable neutrophil-to-lymphocyte ratio, and faster immune cell normalization [26].

These differences in systemic immune tone may modulate not only infection risk but also radiation sensitivity, potentially contributing to the differential acute toxicity profiles observed between surgical approaches.

#### 2.2.4. Periprostatic Adipose Tissue: Histological and Dosimetric Consequences

Minimally invasive prostatectomy often leaves a thicker cuff of periprostatic adipose tissue. This depot is rich in IL-6, IL-8 and leptin, and is intrinsically hypoxic and pro-inflammatory [27]; irradiated adipose tissue developed oxidative stress and reduced adiponectin secretion, changes that may aggravate normal-tissue toxicity [28]. Conversely, preserved fat can provide a few millimeters of natural separation between the prostate bed and rectum; published series, however, suggest that this geometric effect is modest and does not significantly alter rectal dose–volume parameters [29]. Whether the limited spacer benefit outweighs the pro-inflammatory biology of periprostatic adipose tissue is presently unknown and merits dedicated prospective investigation.

#### 2.2.5. Interaction Between Surgical Technique, ADT, and Radiation Toxicity

Androgen deprivation therapy (ADT) plays a complex role in modulating RT toxicity. In primary RT, ADT shrinks prostate volume and improves rectal dosimetry, reducing GI toxicity. However, in the salvage setting post-prostatectomy, the volume reduction benefit is minimal.

Studies such as GETUG-AFU 16 [30] and RTOG 9601 [31] suggest that short-term ADT improves cancer control in salvage RT but does not significantly impact acute toxicity. Biological effects of ADT on fibrosis remain under study: while ADT may modulate TGF-β activity, its influence appears subtle and unrelated to surgical technique.

In summary, the combined effect of open surgery and wider PTV margins likely exacerbates radiation toxicity through a biologically sensitized tissue environment. Conversely, the reduced inflammatory and fibrotic response following minimally invasive surgery may confer relative protection, explaining the observed clinical benefit in toxicity outcomes (Appendix A, Appendix A).

## 3. Toxicity Profile of Post-Prostatectomy Radiotherapy Delivered with Modern Techniques

### 3.1. Impact of Advanced Radiotherapy Techniques on Post-Prostatectomy Toxicity

The introduction of intensity-modulated RT (IMRT), volumetric-modulated arc therapy (VMAT) and daily image guided RT (IGRT) has markedly improved the tolerability of postoperative treatment for prostate cancer. One of the earliest comparative series showed that IMRT reduces acute toxicity even when whole-pelvis fields are required: Among 172 adjuvant- or salvage-treated patients, Alongi et al. [32] reduced grade ≥ 2 GI events from 22% with three-dimensional conformal RT (3D-CRT) to 7% with IMRT and virtually eliminated treatment interruptions. The Memorial Sloan Kettering experience confirmed that these dosimetric gains persist after dose escalation. In fact, Goenka et al. [33] reported that high-dose (median 70 Gy) IMRT lowered five-year late grade ≥ 2 GI toxicity from 10% (3D-CRT) to 1% without increasing GU complications. A multicenter comparison [34] subsequently showed that escalating the salvage-bed dose from 64 Gy (3D-CRT) to 70 Gy (IMRT) maintained biochemical control while keeping late grade ≥ 2 GU-GI toxicity ≤ 5%.

Pairing IMRT with daily IGRT further enhances precision. In a prospective high-risk cohort receiving image-guided IMRT to the whole pelvis, Byun et al. [35] observed only one acute grade 3 GU event (0.6%) and <3% late grade 3 toxicity overall, underscoring the safety of modern IGRT-IMRT in the salvage setting. A 2018 systematic review [36] confirmed that advanced techniques (IMRT, VMAT and IGRT) consistently lower both acute and chronic bowel–bladder morbidity after prostatectomy.

Crucially, these benefits persist even when treatment intensity rises. The phase-III NRG Oncology/RTOG 0534 (SPPORT) trial [37], delivered entirely with IMRT, showed that adding pelvic-node irradiation and short-term androgen-deprivation therapy improved freedom-from-progression while keeping grade ≥ 3 adverse events below 3%.

The next leap is real-time magnetic-resonance-guided adaptation. In a first Japanese series using an MR-linac adaptive workflow [38], the authors reported 69% 5-year biochemical control with no grade ≥ 3 late events, hinting that on-table adaptation may push toxicity even lower.

Collectively, these data illustrate a clear trajectory: each technological refinement (IMRT/VMAT, IGRT, and now on-line adaptation) has incrementally improved the therapeutic ratio of post-prostatectomy RT, achieving durable cancer control with steadily diminishing GU and GI side effects.

### 3.2. Radiation-Induced Tissue Injury: Mechanisms of Cellular and Microenvironmental Damage

RT, while a cornerstone of oncological treatment, inevitably exposes normal tissues to ionizing radiation, eliciting a cascade of biological responses that contribute to both acute and chronic toxicities. The initial event is typically DNA damage, especially double-strand breaks, which represent the most lethal form of injury. These are primarily repaired through homologous recombination and non-homologous end joining, but misrepair or irreparable damage can trigger apoptosis or senescence, especially in proliferative tissues [39]. In addition to direct DNA lesions, oxidative stress induced by ionizing radiation leads to the generation of reactive oxygen and nitrogen species (ROS/RNS). These free radicals cause persistent oxidative damage, mitochondrial dysfunction, and upregulation of enzymes such as NADPH oxidase and cyclooxygenases, which perpetuate cellular injury [40].

At the tissue level, epigenetic alterations and oxidative stress collaborate to dysregulate gene expression and impair regenerative processes. Moreover, radiation disrupts endothelial integrity, promoting inflammation and vascular rarefaction, which exacerbate hypoxia and compromise tissue perfusion [41]. One of the key mediators of the fibrogenic response is transforming growth factor β (TGF-β), which is upregulated following radiation and orchestrates the activation of fibroblasts into myofibroblasts. This process results in extracellular matrix deposition, fibrosis, and long-term tissue stiffening [42,43]. These chronic changes underlie clinical complications such as urethral strictures, decreased bladder compliance, and rectal fibrosis in pelvic RT settings.

Importantly, not all tissues respond equally: the location and functional status of the irradiated organ modulate susceptibility to injury. Advances such as IMRT reduce off-target toxicity, yet inter-individual variability remains significant [44]. Lastly, bystander effects, in which irradiated cells release signals that damage nearby non-irradiated cells, contribute to tissue-level dysfunction and inflammation, amplifying the impact of radiation beyond the targeted volume [45].

### 3.3. Site-Specific Sequelae in Pelvic Organs at Risk

Within the pelvis, the urinary bladder and anterior rectal wall display distinct but inter-related patterns of radiation injury. In the bladder, ionizing radiation denudes the urothelial barrier, incites endothelial loss and free-radical-mediated microvascular obliteration, and triggers a sustained cytokine milieu (IL-6, IL-8, TGF-β1) that drives fibroblast-to-myofibroblast transition. Clinically this evolves from acute hemorrhagic cystitis to late-phase fibrosis, reduced compliance, and urgency–frequency syndromes. Emerging data highlight stromal stem-cell depletion and aberrant angiogenic repair as central mechanisms, providing the rationale for regenerative approaches (e.g., mesenchymal stem-cell or amniotic-membrane instillation) and for hyperbaric oxygen to re-oxygenate hypoxic bladder walls. Parallel translational studies show that germ-line variants in DNA repair and oxidative stress pathways modulate bladder radio-responsiveness, pre-disposing certain patients to severe cystitis or, conversely, to radio-resistant tumors clones that escape local control [46,47].

Radiation damage to the rectum is dominated by matrix remodeling and vascular dysfunction. Early endothelial apoptosis elevates matrix metalloproteinases-2/-9, degrading the submucosal collagen scaffold and disturbing tight-junction integrity; the result is mucosal sloughing, diarrhea and bleeding. Persistent metalloproteinase activity, compounded by progressive capillary loss, fosters chronic hypoxia and collagen I/III deposition, culminating in rigid, fibrotic rectal walls and telangiectasia [48]. Beyond host factors, the gut microbiome has emerged as a critical, modifiable determinant of toxicity. Dysbiosis after pelvic RT amplifies mucosal inflammation, whereas selective antibiotic depletion (e.g., vancomycin) or probiotics can attenuate diarrhea, suggesting that microbial metabolites shape epithelial resilience and immune tone [49].

To reflect this tissue-specific biology, separate models were trained in our study for acute GI and acute GU toxicity. Variable selection and regularization were performed independently for each endpoint.

## 4. Surgical Technique as a Determinant of Acute Toxicity After Salvage Radiotherapy

### 4.1. Clinical Evidence from Our Multicenter Study

Details of case selection, variable definition, and machine-learning workflow can be found in our recent ICAROS publication [14]. Here, we briefly recapitulate the key clinical findings relevant to the current discussion. Our study aimed to identify predictors of acute GI and GU toxicity in patients receiving post-prostatectomy salvage RT [14]. A retrospective multicenter cohort of 454 men treated across three Italian institutions was analyzed using advanced machine-learning approaches, specifically Least Absolute Shrinkage and Selection Operator (LASSO) regression and Classification and Regression Trees (CART). The ISUP grade distribution was as follows: I (17.0%), II (18.3%), III (25.8%), IV (18.9%, V (20.0%). Candidate predictors included surgical approach (open vs. minimally invasive), RT modality, CTV-to-PTV margin, lymphadenectomy extent, and the use/type of androgen deprivation therapy (ADT)

No grade 4 toxicity was recorded; grade 3 events were rare (<1%). Moderate (grade ≥ 2) toxicity occurred in 20.9% of patients for GI and 16.1% for GU endpoints.

Surgical approach emerged as the most powerful predictor of acute toxicity in both CART models (Figure 1 and Figure 2):Patients who had undergone open prostatectomy experienced significantly higher rates of GI (41.8%) and GU (35.9%) toxicity than those treated with laparoscopic or robotic surgery (18.9% and 12.2%, respectively).Among open-surgery patients, CTV-to-PTV margins ≥ 10 mm markedly increased the risk (up to 70.4% for both GI and GU).In patients with minimally invasive surgery, toxicity remained low overall, but EQD2 dose, comorbidity burden, and ADT type modestly modulated GI toxicity.In patients undergoing ORP and irradiated with a CTV-PTV margin < 10 mm, the use of a more extensive lymphadenectomy (≥15 resected nodes) was associated with an increase in both GI and GU toxicity.Among men who underwent minimally invasive surgery and received EQD2 < 70 Gy (N = 85), the crude rate of acute GI toxicity was higher with LHRH agonists (18.9%) than in patients without ADT or treated with high-dose bicalutamide. This comparison is unadjusted and based on a small subgroup with heterogeneous agents; it is therefore hypothesis-generating and not part of the primary inferences.

**Figure 1 cancers-17-02685-f001:**
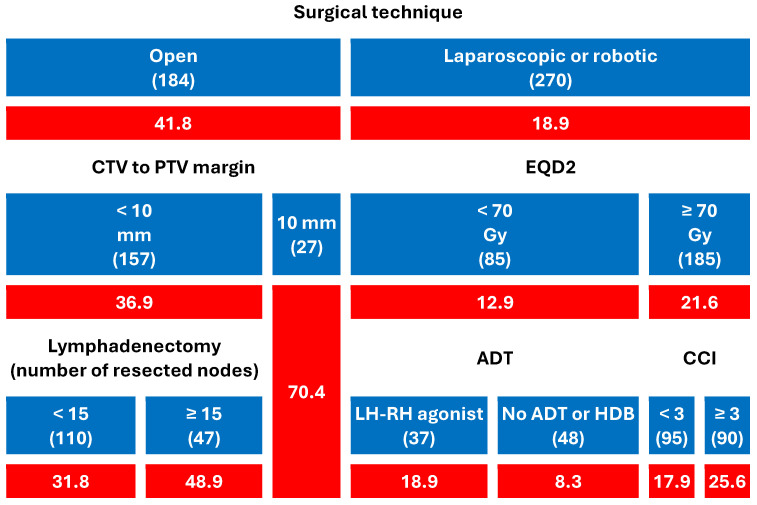
Predictive model of acute gastrointestinal toxicity (grade ≥ 2). The numbers in parentheses represent the patient count within each specific subgroup, whereas the values highlighted on a red background denote the percentages of acute toxicity incidence (ADT: Androgen deprivation therapy, CCI: Charlson Comorbidity Index, CTV: Clinical target volume, EQD2: Equivalent dose in 2 Gy fractions, Gy: Gray, HDB: High-dose bicalutamide, LH-RH agonist: Luteinizing hormone-releasing hormone agonist, PTV: Planning target volume, RT: Radiotherapy).

**Figure 2 cancers-17-02685-f002:**
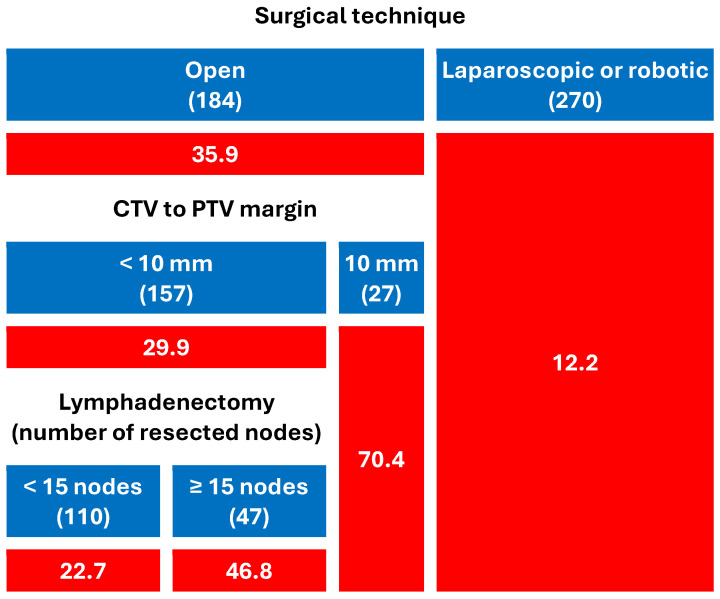
Predictive model of acute genitourinary toxicity (Grade ≥ 2). The numbers in parentheses represent the patient count within each specific subgroup, whereas the values highlighted on a red background denote the percentages of acute toxicity incidence (CTV: Clinical target volume, Gy: Gray, the unit of absorbed radiation dose, PTV: Planning target volume).

In decision-tree (CART) analyses, no partition distinguished laparoscopic from robotic approaches; consequently, these were presented together as the ‘minimally invasive’ group in Figure 1 and Figure 2. Unfortunately, because the original registry lacked granular data on surgeon case-volume, Charlson Comorbidity Index, and Clavien–Dindo complications, residual confounding cannot be excluded; multicenter validations that prospectively collect these variables are encouraged. In particular, a clear limitation is the absence of surgeon and center volume data. Volume–outcome relationships are well described in oncologic surgery [50,51,52,53]), and it is plausible that provider caseload modulates operative trauma, healing biology, and, ultimately, radiation tolerance. Because ICAROS is a radiotherapy registry receiving referrals from urologists with variable caseloads, residual confounding by volume cannot be excluded. Future multicenter efforts will prospectively collect caseload metrics to address this point

Predictive modelling focused on binary acute toxicity outcomes using LASSO logistic regression; time-to-event analyses for oncologic endpoints were not performed because of limited follow-up maturity and completeness for recurrence timing.

### 4.2. Why Surgical Approach Dominated Every Other Predictor of Acute Toxicity

Our machine-learning models disclosed a clear, reproducible signal: after salvage RT, men who had undergone minimally invasive surgery developed roughly half the rate of ≥grade 2 GI and GU events observed after ORP, even when total dose, fractionation, nodal field and beam-delivery technique were identical.

Robotic and pure laparoscopic approaches use smaller ports, shorter ischemia times and far less traction than ORP, producing measurably lower peaks of interleukin-6 and C-reactive protein during the first 24–48 h after surgery [54,55]. Histology also shows that robotic resection causes less capsular incision and preserves a thicker rim of peri-prostatic fat, a natural “spacer” that may later shield bladder and rectum from high-dose radiation [26].

### 4.3. CTV–PTV Margin: A Biological Amplifier

Our CART model shows that in ORP patients the combination of open surgery and 10 mm margin propelled ≥ G2 toxicity to ≈70%, whereas reducing the margin to <10 mm halved that risk, an “amplifier” that converts modest biological vulnerability into major clinical toxicity. In mini-invasive prostatectomy the tighter margin was already standard, so no additional gain was detectable.

### 4.4. Why Beam-Delivery Platform (IMRT vs. 3D-CRT) Looked Neutral

At first glance, it is counter-intuitive that conformal and inverse-planned regimens produced similar acute toxicity. Two considerations reconcile the finding:Surgical/biological dominance. The “surgical imprint” (Section 3.1) and the associated margin size (Section 3.2) together determine how much vulnerable tissue enters the high-dose region; once that volume is set, further modulation of fluence (IMRT vs. 3D-CRT) contributes a smaller incremental change, too small to surface in LASSO selection.Tight contemporary constraints. In our centers, rectum and bladder dose-volume constraints were respected in almost every plan, regardless of technique. When constraints are uniformly achieved, technique differences become statistically less relevant.

This does not imply that IMRT is useless; rather, in a context where both techniques already “pass” constraints, the pre-existing tissue condition dictated the outcome more forcefully than how the dose was painted.

### 4.5. An Integrated Mechanistic Model

Open surgery establishes a hypoxic, cytokine-rich and partially fibrotic pelvic bed. Applying a large margin then sweeps this sensitized tissue into the high-dose region. Radiation activates canonical injury cascades, DNA double-strand breaks, a surge of reactive oxygen species (ROS) and endothelial apoptosis [40,44]. These, in turn, amplify TGF-β signaling and chronic oxidative stress [56], impairing vascular repair and sustaining mucosal injury in the bladder and rectum. The clinical result is the markedly higher incidence of acute GU/GI toxicity observed in patients who combine an open surgical history with wider planning margins.

## 5. Discussion

### 5.1. Unexpected Predictive Power of Surgical Technique

This study identified a previously unrecognized and biologically plausible relationship between the surgical approach used for radical prostatectomy and the risk of acute GI and GU toxicity following postoperative RT. Among the various clinical and technical variables examined, including radiation dose, technique, fractionation, margin size, and nodal field coverage, it was the surgical access, open versus minimally invasive, that most powerfully predicted acute toxicity. Patients who had undergone laparoscopic or robotic prostatectomy experienced roughly half the rate of ≥grade 2 toxicity compared to those treated with ORP, irrespective of the RT modality. This association persisted across model iterations and was further magnified by the use of larger CTV-to-PTV margins, particularly in the open surgery subgroup.

### 5.2. A Novel and Biologically Grounded Hypothesis

To our knowledge, this is the first report to demonstrate such an effect of surgical technique on the biological response to subsequent radiation. The novelty of this finding lies not only in its statistical robustness, but in its clinical implications and conceptual resonance. While open and minimally invasive surgery are often compared in terms of perioperative morbidity, continence recovery, and oncologic control, they have not previously been evaluated through the lens of how they may influence the radiosensitivity of the postoperative tissue environment. Our findings suggest that surgical trauma is not merely a transient perioperative insult [54,55], but leaves a lasting imprint, shaping vascular integrity, cytokine release, tissue oxygenation, and fibrosis [42,43,56], all of which may modulate the vulnerability of pelvic tissues to radiation injury [40,44]. Prospective studies incorporating paired tissue and liquid-biopsy sampling, e.g., longitudinal IL-6/TGF-β panels, hypoxia-specific PET tracers, and collagen-crosslink assays, are now warranted to test the biological model described in this Perspective.

### 5.3. Future Validation Strategies

Because institutional dose-constraint philosophies and surgical referral pathways differ worldwide, the present hypothesis should be tested in multinational collaborations, ideally under a common contouring atlas and harmonized QA program. Pre-existing platforms such as the European POSTOP-RT consortium could provide the needed infrastructure. Moreover, we have to consider some limitations of our study. Health-economics endpoints (operative costs, length of stay, time to return to work) were not collected in the ICAROS registry. Furthermore, the registry did not capture the intraoperative energy modality for tissue detachment in open cases. Because energy use may influence the breadth of thermal injury and the pro-fibrotic milieu, residual confounding cannot be excluded. Future prospective datasets will include standardized recording of energy modality and hemostasis technique to assess their association with toxicity after salvage RT. Finally, lymphadenectomy template and nodal yield were not recorded; thus, potential interactions with periprostatic fat handling could not be assessed.

### 5.4. Planning Margins and Tissue Susceptibility

These insights prompt a reconsideration of RT planning strategies. In particular, the routine adoption of wider PTV margins (e.g., 10 mm) in patients with a history of open surgery may inadvertently amplify toxicity by irradiating tissue rendered more susceptible by surgical trauma. This “amplifier effect” was evident in our model, where the combination of open surgery and a 10 mm margin was associated with the highest rates of acute toxicity. Conversely, patients with minimally invasive surgery were often treated with tighter margins, which likely limited radiation exposure to sensitive regions of the bladder and rectum. It is noteworthy that the mode of radiation delivery, whether 3D conformal or IMRT/VMAT, did not significantly impact outcomes, suggesting that the biological context into which radiation is delivered may matter more than the method of dose modulation, at least when modern planning constraints are uniformly met.

### 5.5. Clinical Implications for Risk Stratification and Planning

Even with contemporary image guidance and planning, irradiated tissues may retain long-lasting fibro-inflammatory changes, which modern techniques can mitigate but not fully reverse. From a clinical standpoint, our observations underscore the need to integrate surgical history into toxicity risk assessments. Patients who underwent open prostatectomy, particularly those who also received extended lymph node dissection, may constitute a high-risk group for acute GI and GU side effects. Such individuals could benefit from early counseling, more vigilant symptom monitoring, and prophylactic interventions during RT. Equally important is the potential for margin individualization: when image guidance is available, smaller margins might safely be adopted in patients at higher risk of toxicity, thereby reducing the irradiated volume and potentially mitigating adverse effects. More specifically, for men treated with open surgery, we recommend a CTV-to-PTV margin < 10 mm (≈6–8 mm based on IGRT system available), whereas margins can be individualized case-by-case in minimally invasive cohorts based on each department motion-management precision.

### 5.6. A Possible “Natural Spacer” Effect of Minimally Invasive Surgery

An additional, previously unconsidered explanation for the observed differential toxicity lies in the anatomical preservation achieved during minimally invasive procedures. Histological studies have shown that robotic resection is associated with less capsular incision and a thicker rim of preserved periprostatic adipose tissue [26]. This residual fat may act as a natural “spacer,” physically separating the prostate bed from adjacent organs such as the rectum and bladder, thus attenuating off-target radiation exposure. Although speculative, this anatomical buffer could contribute to the observed lower toxicity rates and warrants future dosimetric and imaging-based investigation.

## 6. Conclusions

In conclusion, our results suggest that surgical technique exerts a long-lasting biological influence on the tissues subsequently exposed to radiation. Minimally invasive prostatectomy, by limiting vascular and connective tissue injury, appears to leave behind a more resilient pelvic environment, better oxygenated, less inflamed, and more adaptable to the stress of RT. These findings support a more integrated view of prostate cancer care, where surgery and RT are not isolated steps but interact dynamically through shared tissue pathways. Recognizing and harnessing this interplay may help us refine treatment strategies and reduce toxicity for the next generation of patients. Future prospective studies, ideally multinational and biomarker-based, are now required to validate and extend the hypotheses presented in this Perspective.

The implications of these findings extend beyond immediate clinical practice. They point toward new directions for translational research. Prospective studies using functional imaging or fibrosis-sensitive MRI could help characterize the microenvironment of the postoperative bed and validate the biological model proposed here. Tissue sampling and biomarker analyses may further elucidate the molecular signatures of radiation sensitivity as conditioned by surgical trauma. Moreover, the development of radiogenomic models that incorporate surgical variables could enhance our ability to predict and personalize toxicity risk.

## Data Availability

The data of this study are available from the corresponding author upon reasonable request to the corresponding author.

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
