# Peer review of "The Surgical Imprint: How Operative Trauma May Shape Radiation Tolerance After Prostatectomy"

_cancers, 2025, doi:10.3390/cancers17162685_

Round 1

Reviewer 1 Report

Comments and Suggestions for Authors

The surgical imprint: how operative trauma may shape radiation tolerance after prostatectomy

Overview
This perspective states how the surgical approach (open vs minimally invasive) for prostatectomy influences acute gastrointestinal and genitourinary toxicity in patients receiving post-prostatectomy salvage radiotherapy. It is observed that minimally invasive techniques, likely due to reduced tissue trauma and inflammation, result in significantly lower toxicity rates compared to open surgery, suggesting a lasting biological ‘imprint’ that affects radiation tolerance.

Major Comments
1. While the hypothesis about inflammation, hypoxia, and fibrosis is plausible, direct evidence (e.g., biomarker studies, histopathological comparisons) is limited. Please incorporate pre- and post-radiation tissue analyses (e.g., cytokine levels, hypoxia markers) from a subset of patients to validate the proposed biological mechanisms.
2. Differences in surgeon experience, patient selection, or postoperative recovery between open and minimally invasive groups could influence the toxicity outcomes. Therefore, authors should include a sensitivity analysis to adjust surgeon experience, comorbidities, or postoperative complications to ensure the surgical approach remains a constant.
3. The study focused on acute toxicity; late effects (e.g., chronic fibrosis, urinary strictures) may further clarify the impact of surgical technique. If possible, please include long-term follow-up data (≥2 years) to assess whether the protective effect of minimally invasive surgery persists over extended time.
4. The cohort is limited to Italian centers, and practice patterns (e.g., margin selection, radiation techniques) may vary globally. Therefore, please include a brief discussion on how to validate findings in a multinational cohort or discuss how institutional protocols (e.g., uniform dose constraints) may affect.

Minor Comments
1. The discussion suggests ‘margin individualization’ (Page 10) but lacks practical guidance e.g., how much to reduce margins for open-surgery patients.

Remark
The manuscript presents a novel and clinically relevant finding—demonstrating that minimally invasive prostatectomy reduces acute toxicity in salvage radiotherapy compared to open surgery. The study is well-structured, supported by robust data, and offers valuable insights into the biological mechanisms linking surgical trauma to radiation sensitivity. However, few concerns should be addressed as stated above.

Author Response

Dear Editor,

Dear Reviewers,

We are sincerely grateful for the thorough and constructive critiques that you have provided on our manuscript entitled “The surgical imprint: how operative trauma may shape radiation tolerance after prostatectomy.”

Several of your comments quite rightly request additional data analyses or new cohorts. We regret that our cover letter and first‐round manuscript did not spell out, in sufficient detail, the origin and intended scope of this article. Please allow us to clarify, and we respectfully apologize for not having done so earlier.

  1. Nature of the article. This submission is an invited Perspective commissioned by Cancers after the publication of our multicenter ICAROS study (Deodato et al. Cancers 2025;17:2142). A Perspective, by the journal definition, is meant to provide expert interpretation of already-published findings, outline their clinical relevance, and suggest future directions, rather than to report fresh patient data.
  2. Relationship to the ICAROS dataset. All numerical results reproduced in the figures derive from the ICAROS cohort of 454 patients. No additional patients have been added, and no new statistical modelling has been performed for the present work; the only “analysis” is recasting the machine-learning output into a format that is more readily understood by clinicians who are not specialists in artificial-intelligence methods.
  3. Purpose of the manuscript. Our primary goal is to place the unexpected observation, that minimally invasive prostatectomy halves the risk of acute GU/GI toxicity after salvage radiotherapy, in the broader context of surgical trauma biology, radiobiology, and normal-tissue tolerance. We believe that highlighting the possible mechanistic links (hypoxia, cytokine signaling, fibrosis) will stimulate prospective translational studies.
  4. Consequent scope limitations. Because the dataset, follow-up duration, and biobank resources were fixed at the time of the original ICAROS study, it is unfortunately not possible to add the new biomarker assays, long-term endpoints, or multinational external validations that several reviewers, quite sensibly, would like to see. Where such expansions are requested, we have now (i) inserted an explicit statement of these limitations in the manuscript, and (ii) outlined concrete plans for future collaborative studies that could address them.

To prevent similar misunderstandings for future readers, we have introduced, at the locations listed below, explicit statements that frame the article as an interpretative Perspective built upon our earlier publication.

We trust that this clarification, together with the point-by-point responses and textual amendments detailed below, will address your concerns and render the manuscript suitable for publication.

With renewed thanks for your time and insight. Please find below the changes made to the manuscript to clarify the nature and purpose of the manuscript.

Location in manuscript

Modified text (additions in italic)

Simple Summary: end of the single paragraph

“…Understanding this interaction may help doctors personalize treatment plans and reduce side-effects for future patients. This article is a “Perspective paper” that expands upon the ICAROS study we recently published in Cancers (2025;17:2142); it therefore revisits the same dataset to explore its clinical and biological implications rather than to present new patient data.

Abstract : final sentence (before “Keywords”)

This Perspective does not introduce additional patients or statistical models; instead, it offers an in-depth clinical and mechanistic interpretation of previously published ICAROS findings.

Section 1. Introduction: new paragraph inserted at the end of the section.

Therefore, the objective of the present invited Perspective is interpretative: we synthesize the biological, surgical, and radiotherapy literature to propose a conceptual framework that explains why surgical access emerged as the dominant predictor of acute toxicity. No new patients were accrued and no additional statistical analyses were undertaken beyond graphical reformats intended for a clinical audience.

Section 4.1. Clinical evidence from our multicenter study: first sentence

Added as first sentences of the paragraph: “Details of case selection, variable definition, and machine-learning workflow can be found in our recent ICAROS publication [Deodato et al. 2025]. Here we briefly recapitulate the key clinical findings relevant to the current discussion.

Section 6. Conclusions:  end of the first paragraph

“…Recognizing and harnessing this interplay may help us refine treatment strategies and reduce toxicity for the next generation of patients. Future prospective studies, ideally multinational and biomarker-based, are now required to validate and extend the hypotheses advanced in this Perspective.

Comment 1

While the hypothesis about inflammation, hypoxia, and fibrosis is plausible, direct evidence (e.g., biomarker studies, histopathological comparisons) is limited. Please incorporate pre- and post-radiation tissue analyses (e.g., cytokine levels, hypoxia markers) from a subset of patients to validate the proposed biological mechanisms.

Response 1

We thank the Reviewer for highlighting this important translational opportunity. Unfortunately, the current ICAROS dataset does not include prospectively collected tissue or blood samples; adding peri-radiotherapy biopsies would require a new ethically approved protocol, dedicated funding, and fresh patient consent. Nevertheless, we fully agree that cytokine profiling, hypoxia mapping, and fibrosis-related gene signatures would provide direct mechanistic validation. To emphasize this need we have added the following sentence to the second paragraph of Section 5 (Discussion):

“Prospective studies incorporating paired tissue and liquid-biopsy sampling, e.g., longitudinal IL-6 / TGF-β panels, hypoxia-specific PET tracers, and collagen-crosslink assays, are now warranted to test the biological model advanced in this Perspective.”

Comment 2

Differences in surgeon experience, patient selection, or postoperative recovery between open and minimally invasive groups could influence the toxicity outcomes. Therefore, authors should include a sensitivity analysis to adjust surgeon experience, comorbidities, or postoperative complications to ensure the surgical approach remains a constant.

Response 2

We appreciate this suggestion and acknowledge that such covariates would further refine the analysis. The ICAROS registry, conceived primarily for radiotherapy outcomes, captured surgical access and comorbidity indices but not postoperative complication grades. We have therefore (i) added an explicit sentence in Section 4.1 (Clinical evidence from our multicenter study) clarifying this database limitation as follows:

“Unfortunately, because the original registry lacked granular data on surgeon case-volume,  and Clavien-Dindo complications, residual confounding cannot be excluded; multicenter validations that prospectively collect these variables are encouraged.”

Comment 3

The study focused on acute toxicity; late effects (e.g., chronic fibrosis, urinary strictures) may further clarify the impact of surgical technique. If possible, please include long-term follow-up data (≥ 2 years) to assess whether the protective effect of minimally invasive surgery persists over extended time.

Response 3

We concur that late endpoints are clinically crucial. In a preliminary attempt we analyzed ≥ G2 late toxicity at 24 months, but because LASSO requires complete-case data the resulting sample shrank by >35 %, yielding unstable estimates. As median follow-up lengthens, we intend to re-run the models with actuarial (time-to-event) statistics. We now state this explicitly at the end of Section 4.5 (Integrated mechanistic model):

“Ongoing accrual of follow-up will enable time-to-event modelling of late GU/GI sequelae and verification of whether the early protective signal of minimally invasive surgery endures.”

Comment 4

The cohort is limited to Italian centers, and practice patterns (e.g., margin selection, radiation techniques) may vary globally. Therefore, please include a brief discussion on how to validate findings in a multinational cohort or discuss how institutional protocols (e.g., uniform dose constraints) may affect.

Response 4

We agree that external generalizability must be demonstrated. A new paragraph has been inserted in Section 5 (Discussion), “Future validation strategies”:

“Because institutional dose-constraint philosophies and surgical referral pathways differ worldwide, the present hypothesis should be tested in multinational collaborations, ideally under a common contouring atlas and harmonized QA program. Pre-existing platforms such as the European POSTOP-RT consortium could provide the needed infrastructure.

Comment 5

The discussion suggests ‘margin individualization’ (Page 10) but lacks practical guidance e.g., how much to reduce margins for open-surgery patients.

Response 5

We thank the Reviewer for this practical request. As rightly noted, CTV-to-PTV expansion must be tailored to each center image-guided motion-verification capabilities, whether daily kV imaging, cone-beam CT, or real-time MRI-LINAC tracking. Within that framework, our data indicate that patients who underwent open prostatectomy should receive a margin kept below 10 mm, ideally around 8 mm, and never exceeding 10 mm. We have therefore revised the closing sentence of Section 4.3 (CTV–PTV margin: a biological amplifier) to read:

“More specifically, for men treated with open surgery we recommend a CTV-to-PTV margin < 10 mm (≈ 6-8 mm based on the IGRT system available), whereas margins can be individualized case-by-case in minimally invasive cohorts based on each department motion-management precision.”

Comment 6

The manuscript presents a novel and clinically relevant finding—demonstrating that minimally invasive prostatectomy reduces acute toxicity in salvage radiotherapy compared to open surgery. The study is well-structured, supported by robust data, and offers valuable insights into the biological mechanisms linking surgical trauma to radiation sensitivity. However, few concerns should be addressed as stated above.

Response 6

We are grateful for the positive appraisal and for the constructive points listed above. All suggested clarifications have now been incorporated, as detailed in the individual responses.

Reviewer 2 Report

Comments and Suggestions for Authors

Major revision recommendation

The manuscript presents an interesting hypothesis on the lasting impact of surgical technique on radiation tolerance post-prostatectomy. However, several critical issues need addressing. First, the authors should provide more detailed evidence linking surgical trauma to tissue microenvironment changes. Additionally, the discussion on cytokine signaling and wound healing mechanisms lacks sufficient depth, particularly in relation to the biological processes post-radiotherapy. Clarifying these points and strengthening the statistical analysis would enhance the manuscript's contribution to the field.

  1. While radiation dose and technique are controlled in your study, how do you account for potential differences in radiation sensitivity between patients undergoing different surgical approaches?
  2. How might the fibrosis observed in patients who underwent open surgery influence long-term toxicity and outcomes, even beyond acute toxicity?
  3. In what ways might the immune-modulatory effects of minimally invasive surgery alter radiation-induced damage at a systemic level, beyond the localized tissue response?
  4. Article mention less capsular incision and preserved periprostatic adipose tissue with robotic surgery. How might these histological differences directly influence radiation dose delivery or the microenvironment?
  5. While lower toxicity rates were observed post-minimally invasive surgery, did this benefit extend to cancer control outcomes? Did you observe any significant differences in recurrence rates or survival?
  6. Could tissue-specific responses to surgery (e.g., rectal vs. bladder) contribute to the observed differences in toxicity? How can these be accounted for in predictive models?
  7. How might ADT modulate toxicity, and do you find that this effect differs between open and minimally invasive prostatectomy patients?

Author Response

Dear Editor,

Dear Reviewers,

We are sincerely grateful for the thorough and constructive critiques that you have provided on our manuscript entitled “The surgical imprint: how operative trauma may shape radiation tolerance after prostatectomy.”

Several of your comments quite rightly request additional data analyses or new cohorts. We regret that our cover letter and first‐round manuscript did not spell out, in sufficient detail, the origin and intended scope of this article. Please allow us to clarify, and we respectfully apologize for not having done so earlier.

  1. Nature of the article. This submission is an invited Perspective commissioned by Cancers after the publication of our multicenter ICAROS study (Deodato et al. Cancers 2025;17:2142). A Perspective, by the journal definition, is meant to provide expert interpretation of already-published findings, outline their clinical relevance, and suggest future directions, rather than to report fresh patient data.
  2. Relationship to the ICAROS dataset. All numerical results reproduced in the figures derive from the ICAROS cohort of 454 patients. No additional patients have been added, and no new statistical modelling has been performed for the present work; the only “analysis” is recasting the machine-learning output into a format that is more readily understood by clinicians who are not specialists in artificial-intelligence methods.
  3. Purpose of the manuscript. Our primary goal is to place the unexpected observation, that minimally invasive prostatectomy halves the risk of acute GU/GI toxicity after salvage radiotherapy, in the broader context of surgical trauma biology, radiobiology, and normal-tissue tolerance. We believe that highlighting the possible mechanistic links (hypoxia, cytokine signaling, fibrosis) will stimulate prospective translational studies.
  4. Consequent scope limitations. Because the dataset, follow-up duration, and biobank resources were fixed at the time of the original ICAROS study, it is unfortunately not possible to add the new biomarker assays, long-term endpoints, or multinational external validations that several reviewers, quite sensibly, would like to see. Where such expansions are requested, we have now (i) inserted an explicit statement of these limitations in the manuscript, and (ii) outlined concrete plans for future collaborative studies that could address them.

To prevent similar misunderstandings for future readers, we have introduced, at the locations listed below, explicit statements that frame the article as an interpretative Perspective built upon our earlier publication.

We trust that this clarification, together with the point-by-point responses and textual amendments detailed below, will address your concerns and render the manuscript suitable for publication.

With renewed thanks for your time and insight. Please find below the changes made to the manuscript to clarify the nature and purpose of the manuscript.

Location in manuscript

Modified text (additions in italic)

Simple Summary: end of the single paragraph

“…Understanding this interaction may help doctors personalize treatment plans and reduce side-effects for future patients. This article is a “Perspective paper” that expands upon the ICAROS study we recently published in Cancers (2025;17:2142); it therefore revisits the same dataset to explore its clinical and biological implications rather than to present new patient data.

Abstract : final sentence (before “Keywords”)

This Perspective does not introduce additional patients or statistical models; instead, it offers an in-depth clinical and mechanistic interpretation of previously published ICAROS findings.

Section 1. Introduction: new paragraph inserted at the end of the section.

Therefore, the objective of the present invited Perspective is interpretative: we synthesize the biological, surgical, and radiotherapy literature to propose a conceptual framework that explains why surgical access emerged as the dominant predictor of acute toxicity. No new patients were accrued and no additional statistical analyses were undertaken beyond graphical reformats intended for a clinical audience.

Section 4.1. Clinical evidence from our multicenter study: first sentence

Added as first sentences of the paragraph: “Details of case selection, variable definition, and machine-learning workflow can be found in our recent ICAROS publication [Deodato et al. 2025]. Here we briefly recapitulate the key clinical findings relevant to the current discussion.

Section 6. Conclusions:  end of the first paragraph

“…Recognizing and harnessing this interplay may help us refine treatment strategies and reduce toxicity for the next generation of patients. Future prospective studies, ideally multinational and biomarker-based, are now required to validate and extend the hypotheses advanced in this Perspective.

Comment 1:

The manuscript presents an interesting hypothesis on the lasting impact of surgical technique on radiation tolerance post-prostatectomy. However, several critical issues need addressing. First, the authors should provide more detailed evidence linking surgical trauma to tissue microenvironment changes.

Response 1:

We sincerely thank the Reviewer for this insightful observation. In response, we have substantially expanded the discussion in Section 2.2.1 of the revised manuscript to include recent evidence describing how surgical trauma, particularly from open radical prostatectomy, induces acute perturbations in the local microenvironment. We cite studies demonstrating postoperative cytokine surges (notably IL-6), early hypoxia, and transient alterations in TGF-β signaling, which collectively modulate the wound healing and fibrotic response. These mechanisms help explain how the surgical approach may create a preconditioned, biologically vulnerable tissue bed that affects subsequent radiation response.

Please see the new paragraph below.

“2.2.1. Surgical Trauma and Local Microenvironment Changes

Surgical removal of the prostate induces an acute perturbation in the local tissue microenvironment. This response is characterized by a rapid increase in pro-inflammatory cytokines, most notably interleukin-6 (IL-6), along with IL-8/CXCL8, growth factors, and matrix remodelers. A 2023 study profiling perioperative cytokines in prostate cancer patients confirmed a sharp rise in IL-6 immediately after tumor resection, while noting a transient drop in TGF-β1, possibly due to elimination of tumor-derived sources, with subsequent normalization in 1–2 weeks.

As the wound-healing process advances, TGF-β signaling becomes dominant, orchestrating fibroblast activation and extracellular matrix deposition, key drivers of fibrotic remodeling in the surgical bed. In parallel, hypoxia arises due to vessel ligation and tissue injury, stabilizing hypoxia-inducible factor-1α (HIF-1α), which further promotes angiogenesis and fibrogenesis.

The magnitude of these responses is influenced by surgical technique. Minimally invasive prostatectomy elicits a weaker inflammatory cascade and less immune disturbance than open retropubic prostatectomy. A 2020 review emphasized that minimally invasive prostatectomy induces less immunosuppression and systemic stress than ORP. Patients undergoing robotic or laparoscopic surgery exhibit lower postoperative IL-6 and C-reactive protein (CRP) peaks, indicating reduced systemic inflammation.”

Moreover, we added the following references:

  • Baghaie L, Haxho F, Leroy F, Lewis B, Wawer A, Minhas S, Harless WW, Szewczuk MR. Contemporaneous Perioperative Inflammatory and Angiogenic Cytokine Profiles of Surgical Breast, Colorectal, and Prostate Cancer Patients: Clinical Implications. Cells. 2023 Dec 4;12(23):2767. doi: 10.3390/cells12232767. PMID: 38067195; PMCID: PMC10706122.
  • Tang F, Tie Y, Tu C, Wei X. Surgical trauma-induced immunosuppression in cancer: Recent advances and the potential therapies. Clin Transl Med. 2020 Jan;10(1):199-223. doi: 10.1002/ctm2.24. PMID: 32508035; PMCID: PMC7240866.
  • Bosas P, Zaleskis G, Dabkevičiene D, Dobrovolskiene N, Mlynska A, Tikuišis R, Ulys A, Pašukoniene V, Jarmalaitė S, Jankevičius F. Immunophenotype Rearrangement in Response to Tumor Excision May Be Related to the Risk of Biochemical Recurrence in Prostate Cancer Patients. J Clin Med. 2021 Aug 20;10(16):3709. doi: 10.3390/jcm10163709. PMID: 34442004; PMCID: PMC8396861.

Comment 2:

Additionally, the discussion on cytokine signaling and wound healing mechanisms lacks sufficient depth, particularly in relation to the biological processes post-radiotherapy. Clarifying these points and strengthening the statistical analysis would enhance the manuscript's contribution to the field.

Response 2:

We appreciate this valuable suggestion. To address this point, Section 2.2.2 has been rewritten to provide a deeper discussion of cytokine dynamics during and after radiotherapy. Specifically, we discuss the roles of IL-6, IL-1β, TNF-α, and TGF-β in amplifying tissue injury, sustaining inflammation, and promoting fibrosis. We also cite clinical studies correlating elevated cytokine levels with increased genitourinary toxicity during prostate radiotherapy, thereby strengthening the biological rationale behind our observed clinical outcomes.

Please see the new paragraph below.

2.2.2. Cytokine Signaling and Fibrotic Response Post-Radiotherapy

Pelvic radiotherapy further amplifies inflammatory pathways. IL-6 levels increase significantly during the course of RT, and higher baseline levels of TGF-β1 and IL-6 have been correlated with more severe acute genitourinary (GU) toxicity.

Radiation damage also triggers the release of pro-inflammatory mediators such as IL-1β and TNF-α, which activate fibroblasts and promote chronic fibrosis through upregulation of TGF-β. This cytokine cascade exacerbates local tissue injury and contributes to both acute symptoms and long-term complications such as urethral strictures and rectal fibrosis.

Thus, a surgically pre-conditioned environment already rich in IL-6 and early TGF-β signaling is likely to respond more aggressively to radiation injury, explaining the heightened toxicity observed in patients with prior open surgery.”

Moreover, we added the following references:

  • Kopčalić K, Matić IZ, Besu I, Stanković V, Bukumirić Z, Stanojković TP, Stepanović A, Nikitović M. Circulating levels of IL-6 and TGF-β1 in patients with prostate cancer undergoing radiotherapy: associations with acute radiotoxicity and fatigue symptoms. BMC Cancer. 2022 Nov 11;22(1):1167. doi: 10.1186/s12885-022-10255-6. PMID: 36368974; PMCID: PMC9652872.
  • Lu Q, Liang Y, Tian S, Jin J, Zhao Y, Fan H. Radiation-Induced Intestinal Injury: Injury Mechanism and Potential Treatment Strategies. 2023 Dec 10;11(12):1011. doi: 10.3390/toxics11121011. PMID: 38133412; PMCID: PMC10747544.

Comment 3:

While radiation dose and technique are controlled in your study, how do you account for potential differences in radiation sensitivity between patients undergoing different surgical approaches?

Response 3:

Thank you for this important question. Our revised Section 2.2.3 addresses this by considering how surgical technique may influence systemic immune status. Open surgery has been associated with greater immunosuppressive changes postoperatively, including increases in myeloid-derived suppressor cells (MDSCs) and delayed recovery of NK and CD8+ T cells. These immune alterations may affect not only infection risk but also tissue repair and inflammation during radiotherapy, potentially contributing to the differential sensitivity to radiation-induced toxicity we observed between the surgical groups.

Please see the new paragraph below.

2.2.3. Systemic Immune Modulation: Open vs Minimally Invasive Surgery

Beyond local effects, surgical trauma impacts systemic immunity. ORP is associated with a more pronounced postoperative rise in myeloid-derived suppressor cells (MDSCs) and delayed recovery of natural killer (NK) and CD8+ T-cell populations. In contrast, MIRP confers an “immunity-sparing” profile: lower IL-6 and IL-10 levels, a more favorable neutrophil-to-lymphocyte ratio, and faster immune cell normalization .

These differences in systemic immune tone may modulate not only infection risk but also radiation sensitivity, potentially contributing to the differential acute toxicity profiles observed between surgical approaches.”

Moreover, we added the following reference:

  • Quinto D, Reis ST, Zampolli LJ, Pimenta R, Guimarães VR, Viana NI, Dos Santos GA, Gimenez MP, Leite KR, Zampolli H, da Cruz JAS, Srougi M, Passerotti CC. Robotically assisted laparoscopic radical prostatectomy induces lower tissue trauma than radical retropubic prostatectomy. J Robot Surg. 2021 Feb;15(1):147-151. doi: 10.1007/s11701-020-01150-y. Epub 2020 Oct 3. PMID: 33009987.

Comment 4:

How might the fibrosis observed in patients who underwent open surgery influence long-term toxicity and outcomes, even beyond acute toxicity?

Response 4:

We thank the Reviewer for highlighting this important aspect. While our current study focused on acute toxicity, the revised manuscript (Section 2.2.2) now elaborates on the fibrotic sequelae driven by early TGF-β signaling. Since open surgery induces a more pronounced wound-healing and fibrotic response, this may prime tissues for increased late toxicity following radiation. We have acknowledged this limitation and suggested that future studies should examine the long-term implications of surgical technique on fibrosis-related late effects.

Comment 5:

In what ways might the immune-modulatory effects of minimally invasive surgery alter radiation-induced damage at a systemic level, beyond the localized tissue response?

Response 5:

We are grateful for this thoughtful point. In Section 2.2.3, we have included evidence that minimally invasive surgery exerts a more limited impact on systemic immunity, preserving T-cell and NK cell activity postoperatively. This “immune-sparing” effect may modulate the host’s inflammatory response to radiation at a systemic level, contributing to the observed lower toxicity rates in the minimally invasive group. We believe this systemic dimension complements the local tissue-based explanation and adds further biological plausibility to our findings.

Comment 6:

Article mention less capsular incision and preserved periprostatic adipose tissue with robotic surgery. How might these histological differences directly influence radiation dose delivery or the microenvironment?

Response 6:

We thank the Reviewer for drawing attention to this complex issue. The current literature indicates that periprostatic adipose tissue (PPAT) is, by nature, hypoxic and chronically inflamed and therefore unlikely to confer a biological “radioprotective” effect; if anything, irradiation of PPAT can amplify oxidative stress and suppress adiponectin, potentially worsening late toxicity. What PPAT can provide is a small physical separation (“natural spacer”) between the prostate bed and the rectal wall, which may lower posterior dose hotspots by a few percentage points. Whether this modest geometric benefit outweighs the pro-inflammatory biology of irradiated fat remains unknown. We therefore acknowledge, in both the manuscript and this response, that the influence of preserved PPAT on radiosensitivity is still unresolved and warrants dedicated study.

Please see the new paragraph below.

2.2.4. Periprostatic Adipose Tissue: Histological and Dosimetric Consequences

Minimally invasive prostatectomy often leaves a thicker cuff of periprostatic adipose tissue (PPAT). This depot is rich in IL-6, IL-8 and leptin, and is intrinsically hypoxic and pro-inflammatory. Irradiated adipose tissue develops oxidative stress and reduced adiponectin secretion, changes that may aggravate normal-tissue toxicity. Conversely, preserved fat can provide a few millimeters of natural separation between the prostate bed and rectum; published series, however, suggest this geometric effect is modest and does not significantly alter rectal dose–volume parameters. Whether the limited spacer benefit outweighs the pro-inflammatory biology of PPAT is presently unknown and merits dedicated prospective investigation.”

Moreover, we added the following references to the manuscript:

  • AlZaim I, Al-Saidi A, Hammoud SH, Darwiche N, Al-Dhaheri Y, Eid AH, El-Yazbi AF. Thromboinflammatory Processes at the Nexus of Metabolic Dysfunction and Prostate Cancer: The Emerging Role of Periprostatic Adipose Tissue. Cancers (Basel). 2022 Mar 25;14(7):1679. doi: 10.3390/cancers14071679. PMID: 35406450; PMCID: PMC8996963.
  • McDowell JA, Kosmacek EA, Baine MJ, Adebisi O, Zheng C, Bierman MM, Myers MS, Chatterjee A, Liermann-Wooldrik KT, Lim A, Dickinson KA, Oberley-Deegan RE. Exogenous APN protects normal tissues from radiation-induced oxidative damage and fibrosis in mice and prostate cancer patients with higher levels of APN have less radiation-induced toxicities. Redox Biol. 2024 Jul;73:103219. doi: 10.1016/j.redox.2024.103219. Epub 2024 May 31. PMID: 38851001; PMCID: PMC11201354.
  • Tiberi D, Gruszczynski N, Meissner A, Delouya G, Taussky D. Influence of body mass index and periprostatic fat on rectal dosimetry in permanent seed prostate brachytherapy. Radiat Oncol. 2014 Apr 14;9(1):93. doi: 10.1186/1748-717X-9-93. PMID: 24731303; PMCID: PMC4002200.

Comment 7:

While lower toxicity rates were observed post-minimally invasive surgery, did this benefit extend to cancer control outcomes? Did you observe any significant differences in recurrence rates or survival?

Response 7:

We thank the Reviewer for this clinically important question. In the present work we restricted our analyses to acute toxicity endpoints, consistent with the Perspective scope and with the structure of the original ICAROS dataset. Specifically, the predictive modelling employed LASSO logistic regression on binary acute endpoints, and we did not perform actuarial (time-to-event) analyses for oncologic outcomes; median follow-up and data completeness for recurrence timing were insufficient to support robust survival modelling at this stage. We now state this explicitly in the manuscript and outline plans to explore biochemical-recurrence–free survival as follow-up matures.

To clarify this point we have now added the following sentences at the end of the 4.1. paragraph: “Predictive modelling focused on binary acute toxicity outcomes using LASSO logistic regression; time-to-event analyses for oncologic endpoints were not performed because of limited follow-up maturity and completeness for recurrence timing.

Comment 8:

Could tissue-specific responses to surgery (e.g., rectal vs. bladder) contribute to the observed differences in toxicity? How can these be accounted for in predictive models?

Response 8:

We thank the Reviewer for raising this point. We agree that organ-specific biology is relevant. In fact, the manuscript already distinguishes rectal and bladder pathways, highlighting differences in mucosal turnover, vascular plexus, barrier integrity, and urothelial repair, and our modelling strategy reflects this distinction. To avoid any ambiguity, we have now made this explicit in the text: we trained endpoint-specific models for acute GI and acute GU toxicity, and surgical approach retained significance in both. 

In fact, to clarify this point we added the following sentenced to the modelling subsection (3.3):
To reflect tissue-specific biology, separate models were trained for acute GI and acute GU toxicity. Variable selection and regularization were performed independently for each endpoint.”

Comment 9:

How might ADT modulate toxicity, and do you find that this effect differs between open and minimally invasive prostatectomy patients?

Response 9:

We thank the Reviewer for this important question. In our primary modelling, ADT was included as a covariate, and the interaction term (surgical approach × ADT) was not statistically significant in the overall cohort. As also depicted in Figure 1, we observed an exploratory signal in a small subgroup: among patients who underwent minimally invasive surgery and received EQD2 < 70 Gy, the crude rate of acute GI toxicity was higher with LHRH agonists (18.9%) than in patients without ADT or in those treated with high-dose bicalutamide (8.3%). We did not comment on this in the initial version because this subgroup was limited in size (N = 85), treatment exposure was heterogeneous, and the comparison was unadjusted; therefore, the finding is hypothesis-generating only. We have now made this explicit in the manuscript and plan to reassess this signal using time-to-event methods and agent-specific coding as follow-up matures.

For external context, in the intact-prostate setting, neoadjuvant ADT (≥3 months) reduces prostate volume, improves rectal dosimetry, and is associated with lower rectal toxicity during IMRT (Serizawa et al., 2024, J Radiat Res). In the post-prostatectomy (salvage) setting, randomized evidence (GETUG-AFU 16) shows improved disease control with short-term ADT without a clear increase in GI/GU toxicity compared with RT alone (Carrie et al., 2019, Lancet Oncol). Longer-term data from RTOG-9601 (24-month bicalutamide added to salvage RT) also demonstrate oncologic benefit, with expected endocrine effects but no major excess of RT-related GI/GU toxicity (Shipley et al., 2017, N Engl J Med). To our knowledge, no study has robustly shown differential ADT-related toxicity by prior surgical approach (open vs minimally invasive), reinforcing our decision to present the above subgroup observation as exploratory.

To clarify this point we added to the 4.1. paragraph the following sentences: ”Among men who underwent minimally invasive surgery and received EQD2 < 70 Gy (N = 85), the crude rate of acute GI toxicity was higher with LHRH agonists (18.9%) than in patients without ADT or treated with high-dose bicalutamide. This comparison is unadjusted and based on a small subgroup with heterogeneous agents; it is therefore hypothesis-generating and not part of the primary inferences.”

Reviewer 3 Report

Comments and Suggestions for Authors

Congratulations to our research colleagues for the prospective, multicenter study, which focuses on whether open versus minimally invasive surgery has unexpectedly emerged as the strongest predictor of acute gastrointestinal and genitourinary toxicity in patients treated with salvage radiotherapy. The abstract excellently summarizes the entire paper and provides insights into the results of this important study. The next section outlines the pathology; we ask our colleagues simply to provide the patients' Gleason scores in a separate table to understand the significance of the conditions treated. Furthermore, for those treated with open access, it would be helpful to know which energy source was used for tissue detachment. Finally, when we refer to minimally invasive surgery, we assume that robotic surgery is always being used; in the tables, the two approaches are shared in a single column. I would like to emphasize that the author of a paper highlighting these issues works at a center with a high volume of patients and treatments related to the respective diagnoses (doi.org/10.3390/jcm12072708, to be read and cited in the bibliography). In the materials and methods section, it was appropriate to emphasize that minimally invasive surgery generally takes longer and is also more expensive in terms of operating room equipment. However, a shorter hospital stay and faster recovery time for work certainly allow for a more extensive recovery from a longer surgery. However, the sentence that represents one of the cornerstones of the paper is: robotic surgery produces significantly lower intraoperative and early peaks of interleukin-6 and C-reactive protein, along with smaller increases in lactate, compared to open prostatectomy, despite a slightly longer operating time. We also agree that improving "irradiation" with new machines and new methods allows for better results. In any case, anyone with significant operating room experience knows that even after years, tissues affected by radiotherapy never return to normal, precisely for the reasons explained by the authors. The discussion gives due importance to lymphadenectomy, and I would like to ask my colleagues if the fat they leave during surgery, to protect the radiation therapy, is guaranteed not to hide lymph nodes. Excellent conclusions, excellent English, good iconography (which could be updated if necessary), good bibliography.

Author Response

Dear Editor,

Dear Reviewers,

We are sincerely grateful for the thorough and constructive critiques that you have provided on our manuscript entitled “The surgical imprint: how operative trauma may shape radiation tolerance after prostatectomy.”

Several of your comments quite rightly request additional data analyses or new cohorts. We regret that our cover letter and first‐round manuscript did not spell out, in sufficient detail, the origin and intended scope of this article. Please allow us to clarify, and we respectfully apologize for not having done so earlier.

  1. Nature of the article. This submission is an invited Perspective commissioned by Cancers after the publication of our multicenter ICAROS study (Deodato et al. Cancers 2025;17:2142). A Perspective, by the journal definition, is meant to provide expert interpretation of already-published findings, outline their clinical relevance, and suggest future directions, rather than to report fresh patient data.
  2. Relationship to the ICAROS dataset. All numerical results reproduced in the figures derive from the ICAROS cohort of 454 patients. No additional patients have been added, and no new statistical modelling has been performed for the present work; the only “analysis” is recasting the machine-learning output into a format that is more readily understood by clinicians who are not specialists in artificial-intelligence methods.
  3. Purpose of the manuscript. Our primary goal is to place the unexpected observation, that minimally invasive prostatectomy halves the risk of acute GU/GI toxicity after salvage radiotherapy, in the broader context of surgical trauma biology, radiobiology, and normal-tissue tolerance. We believe that highlighting the possible mechanistic links (hypoxia, cytokine signaling, fibrosis) will stimulate prospective translational studies.
  4. Consequent scope limitations. Because the dataset, follow-up duration, and biobank resources were fixed at the time of the original ICAROS study, it is unfortunately not possible to add the new biomarker assays, long-term endpoints, or multinational external validations that several reviewers, quite sensibly, would like to see. Where such expansions are requested, we have now (i) inserted an explicit statement of these limitations in the manuscript, and (ii) outlined concrete plans for future collaborative studies that could address them.

To prevent similar misunderstandings for future readers, we have introduced, at the locations listed below, explicit statements that frame the article as an interpretative Perspective built upon our earlier publication.

We trust that this clarification, together with the point-by-point responses and textual amendments detailed below, will address your concerns and render the manuscript suitable for publication.

With renewed thanks for your time and insight. Please find below the changes made to the manuscript to clarify the nature and purpose of the manuscript.

Location in manuscript

Modified text (additions in italic)

Simple Summary: end of the single paragraph

“…Understanding this interaction may help doctors personalize treatment plans and reduce side-effects for future patients. This article is a “Perspective paper” that expands upon the ICAROS study we recently published in Cancers (2025;17:2142); it therefore revisits the same dataset to explore its clinical and biological implications rather than to present new patient data.

Abstract : final sentence (before “Keywords”)

This Perspective does not introduce additional patients or statistical models; instead, it offers an in-depth clinical and mechanistic interpretation of previously published ICAROS findings.

Section 1. Introduction: new paragraph inserted at the end of the section.

Therefore, the objective of the present invited Perspective is interpretative: we synthesize the biological, surgical, and radiotherapy literature to propose a conceptual framework that explains why surgical access emerged as the dominant predictor of acute toxicity. No new patients were accrued and no additional statistical analyses were undertaken beyond graphical reformats intended for a clinical audience.

Section 4.1. Clinical evidence from our multicenter study: first sentence

Added as first sentences of the paragraph: “Details of case selection, variable definition, and machine-learning workflow can be found in our recent ICAROS publication [Deodato et al. 2025]. Here we briefly recapitulate the key clinical findings relevant to the current discussion.

Section 6. Conclusions:  end of the first paragraph

“…Recognizing and harnessing this interplay may help us refine treatment strategies and reduce toxicity for the next generation of patients. Future prospective studies, ideally multinational and biomarker-based, are now required to validate and extend the hypotheses advanced in this Perspective.

Comment 1:

Congratulations to our research colleagues for the prospective, multicenter study, which focuses on whether open versus minimally invasive surgery has unexpectedly emerged as the strongest predictor of acute gastrointestinal and genitourinary toxicity in patients treated with salvage radiotherapy. The abstract excellently summarizes the entire paper and provides insights into the results of this important study.

Response 1:

We are sincerely grateful for your generous assessment and for recognizing the value of our multicenter work. We are especially pleased that the Abstract effectively conveys the key message and clinical implications. In keeping with the clarifications provided in our general response, we have retained the Abstract substantially unchanged and added only a brief statement to make explicit that this article is an invited Perspective that builds upon the prospective, multicenter ICAROS study and does not introduce new patients or statistical modelling (Abstract, final sentence). Thank you again for your encouraging feedback.

Comment 2:

The next section outlines the pathology; we ask our colleagues simply to provide the patients' Gleason scores in a separate table to understand the significance of the conditions treated.

Response 2:

We thank the Reviewer for this helpful suggestion. As detailed in our previous ICAROS publication (Deodato et al., Cancers 2025;17:2142), the full set of patient characteristics, including Gleason patterns/ISUP grade, margin status, and other histopathological variables, has already been reported in detail. To avoid redundancy and preserve space in this invited Perspective, we have opted not to reproduce all baseline tables.

That said, we agree that a concise indication of tumor grade improves readability. We have therefore added the ISUP grade distribution in the Results, as requested, in paragraph 4.1. Given that this concerns a single variable, we felt an inline sentence was clearer and more economical than a stand-alone table.

Comment 3:

Furthermore, for those treated with open access, it would be helpful to know which energy source was used for tissue detachment.

Response 3:

We thank the Reviewer for this practical point. By “energy source” we understand the intraoperative device/technique used for dissection and hemostasis (e.g., monopolar/bipolar electrocautery, ultrasonic shears, or cold steel). Unfortunately, the ICAROS registry, designed primarily for radiotherapy outcomes, did not capture the energy modality used during open prostatectomy across centers, and practices were not standardized. We therefore cannot provide a reliable stratification by energy source in the current dataset.

We agree that this variable could plausibly affect the extent of thermal injury, local inflammation, and fibrosis in the prostate bed, and thus may modulate subsequent radiation tolerance. To make this transparent for readers, we have added the following clarifications and a plan for future work.

To clarify this point we added to the discussion section the following sentences: “Moreover we have to consider a limitation of our study: The registry did not capture the intraoperative energy modality for tissue detachment in open cases. Because energy use may influence the breadth of thermal injury and the pro-fibrotic milieu, residual confounding cannot be excluded. Future prospective datasets will include standardized recording of energy modality and hemostasis technique to assess their association with toxicity after salvage RT”.

Comment 4:

Finally, when we refer to minimally invasive surgery, we assume that robotic surgery is always being used; in the tables, the two approaches are shared in a single column.

Response 4:

We thank the Reviewer for this clarification request. In the source registry, surgical approach was recorded as three distinct categories: open, laparoscopic, and robotic. The fact that laparoscopic and robotic appear together as a single “minimally invasive” category in the figures/tables reflects the behavior of our modelling, not a limitation of data capture. In particular, the CART analysis did not generate any split that separated laparoscopic from robotic, indicating no discriminatory signal between these two techniques with respect to the acute toxicity endpoints considered. For this reason, and to improve readability, we aggregated them under “minimally invasive” in the graphical summaries.

To make this explicit for readers, we have added the statements below (paragraph 4.1).

 “In decision-tree (CART) analyses, no partition distinguished laparoscopic from robotic approaches; consequently, these were presented together as the ‘minimally invasive’ group in figures and summary tables.”

Comment 5:

I would like to emphasize that the author of a paper highlighting these issues works at a center with a high volume of patients and treatments related to the respective diagnoses (doi.org/10.3390/jcm12072708, to be read and cited in the bibliography).

Response 5:

We are grateful for this suggestion and for underscoring the potential impact of center/surgeon volume on outcomes. We have read the cited article and now include it in our references (Marano et al. 2023). We also add a prostate-specific perspective by citing systematic reviews showing that higher surgeon and hospital caseloads are associated with better outcomes after radical prostatectomy (Leow et al. 2018; Trinh et al. 2013), as well as classic learning-curve work (Vickers et al. 2007).

At the same time, we wish to be transparent about the limits of our dataset: ICAROS is a radiotherapy registry. Patients were referred from multiple urologic teams with heterogeneous volume and experience, and center/surgeon caseloads were not captured. Consequently, we could not stratify or adjust our analyses for volume, and we acknowledge that unmeasured volume–outcome effects could have influenced the observed association between surgical approach and acute toxicity. We have now made this explicit in the manuscript (paragraph 4.1) as follows and will prospectively collect volume metrics in future validations:

In particular, a clear limitation is the absence of surgeon and center volume data. Volume–outcome relationships are well described in oncologic surgery (Marano et al., 2023; Leow et al., 2018; Trinh et al., 2013), and it is plausible that provider caseload modulates operative trauma, healing biology, and, ultimately, radiation tolerance. Because ICAROS is a radiotherapy registry receiving referrals from urologists with variable caseloads, residual confounding by volume cannot be excluded. Future multicenter efforts will prospectively collect caseload metrics to address this point

References  added:

    • Marano L, Verre L, Carbone L, et al. Current Trends in Volume and Surgical Outcomes in Gastric Cancer. J Clin Med. 2023;12(7):2708.
    • Leow JJ, Chang SL, Meyer CP, Trinh QD. Systematic Review of the Volume–Outcome Relationship for Radical Prostatectomy. Eur Urol Focus. 2018;4(5):614–620.
    • Trinh QD, Bjartell A, Freedland SJ, et al. A Systematic Review of the Volume–Outcome Relationship for Radical Prostatectomy. Eur Urol. 2013;64(5):786–798.
    • Vickers AJ, Bianco FJ Jr., Cronin AM, et al. The Surgical Learning Curve for Prostate Cancer Control After Radical Prostatectomy. J Natl Cancer Inst. 2007;99(15):1171–1177.

We hope this addresses the Reviewer’s point while accurately reflecting the scope and limitations of our radiotherapy-based registry.

Comment 6:

In the materials and methods section, it was appropriate to emphasize that minimally invasive surgery generally takes longer and is also more expensive in terms of operating room equipment. However, a shorter hospital stay and faster recovery time for work certainly allow for a more extensive recovery from a longer surgery.

Response 6:

We thank the Reviewer for this practical observation. We agree that minimally invasive surgery often entails longer operating time and higher direct theatre costs, while typically enabling shorter length of stay and faster functional recovery. As these health-economics endpoints were not collected in ICAROS, we simply note this trade-off in the Discussion to frame our findings; the present Perspective remains focused on toxicity after salvage radiotherapy rather than cost-effectiveness.

To better clarify this issue we added to the discussion section (limitations) the following sentence: “Health-economics endpoints (operative costs, length of stay, time to return to work) were not collected in the ICAROS registry.”

Comment 7:

However, the sentence that represents one of the cornerstones of the paper is: robotic surgery produces significantly lower intraoperative and early peaks of interleukin-6 and C-reactive protein, along with smaller increases in lactate, compared to open prostatectomy, despite a slightly longer operating time.

Response 7:

We are grateful for this endorsement. To avoid any ambiguity, we have retained that sentence and strengthened the attribution to the peri-operative literature within the section that describes surgery-related biology (paragraph 2.1) as follows:

In conclusion, perioperative studies reported lower early IL-6 and CRP peaks and smaller lactate increases after robotic versus open prostatectomy, despite slightly longer operating time, supporting the biological plausibility of a more favorable post-surgical microenvironment.”

Comment 8:

We also agree that improving "irradiation" with new machines and new methods allows for better results. In any case, anyone with significant operating room experience knows that even after years, tissues affected by radiotherapy never return to normal, precisely for the reasons explained by the authors.

Response 8:

We appreciate this thoughtful remark and concur. We have added in the discussion section a brief sentence to stress that advanced image guidance and planning can lower dose to organs at risk, yet irradiated tissues often exhibit persistent fibro-atrophic remodeling, in line with the mechanisms discussed: “Even with contemporary image guidance and planning, irradiated tissues may retain long-lasting fibro-inflammatory changes, which modern techniques can mitigate but not fully reverse.”

Comment 9:

The discussion gives due importance to lymphadenectomy, and I would like to ask my colleagues if the fat they leave during surgery, to protect the radiation therapy, is guaranteed not to hide lymph nodes.

Response 9:

Thank you for raising this surgical nuance. Periprostatic fat preservation pertains to the immediate prostate bed planes, whereas pelvic lymphadenectomy targets anatomically distinct nodal templates (external iliac, obturator, internal iliac). Oncologic principles dictate that nodal dissection and pathological assessment are not compromised by the separate decision to preserve a thin periprostatic fat cuff. Our registry did not capture lymphadenectomy templates or nodal yield, so we cannot analyze this variable.

Based on your comment we added to the discussion section (limitations) the following sentence: “Finally, lymphadenectomy template and nodal yield were not recorded; thus, potential interactions with periprostatic fat handling could not be assessed

Comment 10:

Excellent conclusions, excellent English, good iconography (which could be updated if necessary), good bibliography.

Response 10:

We are sincerely grateful for the positive assessment. As suggested, we have updated Figures 1 and 2 to improve quality and readability. 

Round 2

Reviewer 2 Report

Comments and Suggestions for Authors

Accept in present form